# Monitoring of Vitamin C Plasma Levels in a Reversible Model of Malabsorption Generated in Mice by Ebulin-f

**DOI:** 10.3390/toxins17070333

**Published:** 2025-06-30

**Authors:** Daniel Arranz-Paraiso, M. Angeles Rojo, Cristina Martin-Sabroso, Manuel Cordoba-Diaz, Tomás Girbés, Manuel Garrosa, Damian Cordoba-Diaz

**Affiliations:** 1Area of Pharmaceutics and Food Technology, Faculty of Pharmacy, Complutense University of Madrid, 28040 Madrid, Spain; sandani_@hotmail.com (D.A.-P.); crmartin@ucm.es (C.M.-S.); mcordoba@ucm.es (M.C.-D.); 2Area of Experimental Sciences, Miguel de Cervantes European University, 47012 Valladolid, Spain; marojo@uemc.es; 3University Institute of Industrial Pharmacy (IUFI), Complutense University of Madrid, 28040 Madrid, Spain; 4Area of Nutrition and Food Sciences, University of Valladolid, 47005 Valladolid, Spain; tomas.girbes@uva.es; 5Area of Histology, Faculty of Medicine and INCYL, University of Valladolid, 47005 Valladolid, Spain

**Keywords:** vitamin C, inflammatory bowel diseases, ebulin-f, RIPs, reversible model, *Sambucus ebulus*, biopharmaceutics, malabsorption

## Abstract

The development of reversible animal models for the study of intestinal pathologies is essential to reduce the number of animals used in research and to better understand disease mechanisms. In this study, we present a reversible model of intestinal malabsorption through the administration of sublethal doses of ebulin-f, a ribosome-inactivating protein, and validate its usefulness by monitoring vitamin C absorption. The scientific community increasingly recognizes the importance of rationalizing experimental designs, optimizing treatment protocols, and minimizing the use of animals in research models. Thus, new methodologies are needed to minimize invasive sampling and to develop reversible animal models that recover physiologically post-study. Such models are essential for in vivo studies of human pathologies. Sublethal doses of ebulin-f (2.5 mg/kg) administered intraperitoneally to female Swiss CD1 mice (*n* = 6 per group) can cause reversible intestinal alterations in the small intestine, which offer the possibility of having a valuable reversible study model of malabsorption for the investigation of this syndrome. To verify whether nutrient absorption is altered, we used vitamin C as a traceable nutrient that can be quantified in the blood. Peripheral blood samples were collected through the retro-orbital area at 30, 80, 120, 180, and 1440 min post-administration, treated with DTT and MPA, and analyzed using a validated UV/Vis–HPLC method to indirectly determine vitamin C absorption by enterocytes. Pharmacokinetic analysis revealed significantly increased vitamin C absorption on days 1 and 3 post-treatment (AUC values of 3.65 × 10^4^ and 7.10 × 10^4^, respectively) compared to control (0.94 × 10^4^), with partial recovery by day 22 (3.27 × 10^4^). Blood concentration profiles indicate that intestinal damage peaks at day 3, followed by significant regeneration by day 22, establishing this as a viable reversible model for inflammatory bowel disease research.

## 1. Introduction

The human digestive system comprises solid and hollow organs, primarily functioning to absorb nutrients from food. The intestinal mucosa consists of two main structures, the epithelium and the lamina propria, organized into protrusions (villi) and invaginations (crypts). The luminal surface of the small intestine epithelium is highly amplified by the microvilli of the enterocytes. These cells contain enzyme families that catalyze the final digestion of polysaccharides and peptides and the luminal hydrolysis of various polymers [1]. Thus, the primary function of the small intestine is to ensure adequate nutrient uptake into the body, and this is carried out through the processes of digestion and absorption. When these functions fail, maldigestion and malabsorption occur, presenting characteristic clinical features.

Drugs and other xenobiotics are primarily absorbed via diffusion or through specific receptors along the gut mucosa. During pharmaceutical development, it should be demonstrated that the active pharmaceutical ingredient is able to “escape” from the dosage form in which it is presented to the body, dissolve in the lumen, and reach therapeutical plasma levels. In this sense, in the preclinical phase of the development, release, dissolution, and absorption studies should be performed in ex vivo, in vitro, in silico, and animal models as similar as possible to the human gastrointestinal tract. These models vary in complexity and applicability to in vivo situations. Recently, Macedo et al. [2] developed an interesting bioprinted 3D model that could be the gold standard model in the near future; however, at present, there is no model that can be used as a complete replacement for malabsorption syndrome [3,4,5].

Malabsorption can be caused by diseases of the small intestine, as well as by diseases of the pancreas, liver, biliary tract, and stomach [6]. Specifically, the etiology of malabsorption includes digestive enzyme deficiency or inactivation [7], decreased bile acid synthesis and/or secretion [8], reduced gastric acid production, intrinsic factor or pancreatic protease deficiency, food intolerances (fructose, carbohydrates) [7], specific nutrient transport defects such as Hartnup disease or cystinuria, and immune-mediated conditions like celiac disease [6]. Symptoms associated with malabsorption are functional and non-specific [9]. Due to the wide etiological variety of malabsorption, the treatment approach is quite complex. Therefore, the development of in vitro, ex vivo, and in silico systems, and especially animal models, is key to research and advancement in the treatment of these diseases [3,4,5]. Recent technological advances in this field include microfluidic intestinal models and advanced organoid culture systems that better recapitulate the physiological conditions of the human intestine [10,11].

To study intestinal diseases, non-reversible models can be generated by administering xenobiotics systemically or directly to the intestine. For example, xenobiotics such as steroids [12], acetic acid [13,14], sodium dextran [14], or trinitrobenzenic acid [15] have been used. Another option is to use genetically modified animals in which chronic diseases develop, but this model has the disadvantage of the time it takes to develop [16,17]. Examples of genetic models are the CH3/HeJBir mice [18,19], which show an alteration at the end of the large intestine, and the SAMP1/Yit and SAMP1/YitFc mice, which develop ileitis at 10 weeks of age [20]. The principles of replacing, reducing, and refining animal use in research [21] underscore the need for reversible animal models and diagnostic techniques. These allow malabsorption syndrome studies without animal sacrifice, representing a significant advancement.

Ribosome-inactivating proteins (RIPs) are a group of cytotoxic proteins widely dis-tributed in eukaryotic and prokaryotic organisms. They contain at least one non-catalytic domain that allows them to reversibly bind to glycoconjugates without causing structural modifications. Although the most recognized activity of RIPs is blocking the biosynthesis of proteins through a specific depurination in the alpha-sarcin loop of rRNA, some of them induce cell apoptosis by other mechanisms such as depurination of DNA. Several clinical trials and preclinical studies have been performed using RIPs (immunotoxins, PEGylated complexes, nano/microcarriers) as antibacterial, antifungal, antimicrobial, antitumor, or antiviral agents. Nevertheless, due to their nonspecific toxicity to the lungs, kidneys, or CNS, immunogenicity, or limited pharmacokinetic properties, RIPs are not yet used as drugs [22,23,24,25,26].

Type II RIPs consist of two peptide chains linked through a disulfide bridge. One of these chains has enzymatic activity and the other one has lectin properties [27,28]. These proteins should be classified into two subgroups: high toxicity (ricin, abrin, volkensin, viscumin, and modeccin) [29] and low toxicity (ebulitins and nigrins) [30,31]. Differences in toxicity depend on their ability to recognize cell membrane receptors (binding lectin to cell membrane galactose residues) [32] and their capacity to induce cell internalization [29].

Previous studies in mice have shown that administration of sublethal concentrations of type II Sambucus ribosome-inactivating proteins (RIPs), namely nigrin b [33,34] and ebulin-f [35,36], can cause reversible intestinal alterations. It has been described that 24 h after i.p. of nigrin b, the crypts of Lieberkühn show apoptotic morphology, an atrophied profile, and a significant reduction of intestinal villi. Likewise, the literature describes that three days after nigrin b administration, the villi of the first portion of the small intestine are able to regenerate with high mitotic activity [33,34] because the intestinal stem cell niche is not completely destroyed, allowing crypt regeneration through functional crosstalk with other cell types such as Paneth cells [34].

Although nigrin b and ebulin-f exhibit similar intestinal toxicity profiles, it has been previously reported that the LD_50_ after i.p. administration differs for both proteins. Since ebulin-f is more toxic, the induction of rapid and pronounced intestinal lesions at lower doses facilitates the investigation of cellular and molecular mechanisms underlying tissue injury. Additionally, our previous research has established that ebulin-f exhibits more selective targeting of intestinal transit-amplifying compartment (TAC) cells with less systemic toxicity [35,36].

Therefore, due to the reversible alteration they cause in the small intestine, it is considered that the use of these RIPs for the generation of a reversible model of malabsorption could be a valuable resource for the investigation of this syndrome. In particular, the similarity between the activity of nigrin b and ebulin-f is the starting point for hypothesizing the new reversible model of ebulin-f. In addition, the use of ebulin-f has the advantage over nigrin-b that, since it is not glycosylated, its immunogenicity will presumably be lower.

To avoid animal sacrifice, it is not only important to develop a reversible model but also to validate a suitable minimally invasive analytical method to evaluate the evolution of the pathology. With this approach, we propose the evaluation of malabsorption syn-drome by monitoring plasma vitamin C levels.

Vitamin C (ascorbic acid) is taken up only by intestinal absorption during food di-gestion. This absorption is mediated by sodium-ascorbate cotransporters (sodium-vitamin C transporters, SVCT), of which there are two isoforms: SVCT1 and SVCT2. The SVCT1 isoform is responsible for active transport from the luminal surface of the gastrointestinal mucosa of the small intestine and for its reabsorption in the renal proximal tubule, thus determining the plasma levels of the vitamin, whereas SVCT2 is ubiquitous and responsible for its bioaccumulation [37,38].

The absorption process is primarily mediated by the sodium-dependent vitamin C transporter 1 (SVCT1), a high-affinity, saturable transporter encoded by the SLC23A1 gene [39]. SVCT1 operates via a sodium-coupled active transport mechanism, exhibiting Michaelis–Menten kinetics with a Km of approximately 0.2–0.3 mM, indicating that absorption efficiency peaks at moderate luminal concentrations but diminishes at higher doses due to transporter saturation [40]. In healthy individuals, oral doses of 200–1000 mg achieve peak plasma concentrations of 60–100 µM within 2–3 h, with bioavailability declining significantly at doses exceeding 1 g due to both transporter limitations and increased renal excretion of unmetabolized ascorbic acid.

The sodium-dependent vitamin C transporter 2 (SVCT2), encoded by SLC23A2, plays a secondary role in intestinal absorption but is more critical for vitamin C uptake in other tissues, such as the brain and kidneys [41]. While active transport via SVCT1 dominates under physiological conditions, passive diffusion of ascorbic acid is minimal due to its ionization at intestinal pH (pKa ~4.2), which renders it negatively charged and poorly permeable across lipid membranes [42]. However, dehydroascorbic acid, the oxidized form of vitamin C, can be absorbed via facilitative glucose transporters (GLUT1 and GLUT3) in the intestinal epithelium. Once inside enterocytes, dehydroascorbic acid is rapidly reduced to ascorbic acid by glutathione-dependent enzymes, contributing to the intracellular pool [42]. This alternative pathway is particularly relevant under conditions of oxidative stress, where dehydroascorbic acid levels may increase, though its overall contribution to absorption remains small compared to SVCT1-mediated transport.

Several factors influence the absorption kinetics of vitamin C. Luminal pH affects SVCT1 activity, with optimal function occurring at slightly acidic to neutral conditions; significant deviations, such as those in gastric or intestinal disorders, can impair transport efficiency. Intestinal transit time also plays a role, as prolonged or accelerated transit may reduce contact time with absorptive surfaces, particularly in the jejunum. Dietary components, notably glucose, compete with dehydroascorbic acid for GLUT transporter binding, potentially reducing absorption of the oxidized form, though this effect is less significant for ascorbic acid itself [42]. Emerging research highlights the role of the gut microbiota in modulating vitamin C bioavailability. Certain microbial species may alter luminal redox conditions, stabilizing ascorbic acid or promoting its oxidation to dehydroascorbic acid, which could affect absorption efficiency [43]. For instance, Lactobacillus and Bifidobacterium species have been shown to enhance ascorbic acid stability in vitro, suggesting a potential indirect effect on bioavailability, though in vivo data remain limited [43].

Pharmacokinetic studies indicate that vitamin C absorption is highly dose-dependent. At low doses (e.g., 15–100 mg), absorption efficiency approaches 80–90%, but it drops to less than 50% at doses above 1 g due to saturation of SVCT1 and increased urinary losses [40]. The half-life of vitamin C in plasma is approximately 8–12 h, with steady-state concentrations tightly regulated by renal reabsorption and excretion mechanisms [39]. These kinetics underscore the importance of frequent, moderate dosing to maintain optimal plasma levels, particularly in conditions where absorption may be compromised.

Recent advances in understanding absorption kinetics have also explored the impact of food matrix effects. For example, co-ingestion of vitamin C with dietary fats or fibers may delay gastric emptying, prolonging intestinal transit and potentially enhancing absorption by extending contact time with SVCT1-expressing enterocytes [44]. Conversely, high-fiber diets may bind ascorbic acid, reducing its bioavailability, though evidence is inconclusive [43,45]. These findings highlight the complexity of vitamin C absorption and the need for further research into dietary and physiological modulators.

At the level of erythrocytes, leukocytes, the blood–brain barrier, and the blood–testicular barrier, glucose-facilitating transporters (GLUTs), such as GLUT1, GLUT2, GLUT3, and GLUT10, responsible for the transport of the oxidized form of vitamin C, DHA, are also found [39,43]. Any alteration in these systems can trigger moderate vitamin C deficiency, with the presence of SVCT1 and SVCT2 being essential to maintain vitamin C homeostasis and ruling out sufficient involvement of GLUT transporters [38]. Pathologies such as inflammatory bowel diseases generate deficient plasma levels of this vitamin [46,47] due to an alteration in the SVCT1 and SVCT2 cotransporters present at the intestinal level. Therefore, the monitoring of vitamin C plasma levels is directly related to intestinal functionality.

This study aims to develop a reversible model of intestinal malabsorption using i.p. administration of sublethal ebulin-f doses. This model allows for evaluating the effectiveness of new drug candidates for the treatment of diseases that damage the intestinal mucosa, such as Crohn’s disease or irritable bowel syndrome. The model will be evaluated by quantifying plasma levels of vitamin C in animals treated with ebulin-f.

## 2. Results

### 2.1. Isolation and Characterization of Ebulin-f

As shown in Figure 1 (left), affinity chromatography was not sufficient to obtain pure ebulin-f from the fresh extract. Nevertheless, with this first isolation, lectins with affinity for galactose were obtained. Aliquots 15–30 were pooled and concentrated in an Amicon^®^ before their elution through a size-exclusion chromatography column. The obtained chromatogram (Figure 1—right) shows two defined peaks but is not completely resolved. To avoid impurities from SELfd, ebulin-f was isolated from aliquots 33 and 34.

Characterization of the activity of ebulin-f was performed by hemagglutination of red blood cells from human blood type O, Rh +, at 24 °C and 4 °C. Ebulin-f was able to agglutinate it at 24 °C with a minimum protein concentration of 37.5 µg/mL and at 4 °C with a minimum concentration of 18.8 µg/mL (Figure 2). These results are in accordance with previous results from our research group.

Purity was demonstrated by two-dimensional electrophoresis in the presence of SDS (SDS-PAGE). As expected, nearby bands were obtained from SELfd and ebulin-f aliquots, which is logical given their approximately equal molecular weights. Samples treated with mercaptoethanol presented two bands due to the dissolution of the disulfide bridges between the two chains of the proteins.

### 2.2. Vitamin C Determination by HPLC/UV-Vis

Table 1 shows the results of the main validation parameters of the method for quantification of vitamin C in plasma according to the ICH Q2A recommendations. The analytical method has proper accuracy, robustness and precision in the linear range from 6–60 μg/mL.

Specificity of the analytical method refers to its ability to accurately identify and quantify vitamin C in the presence of impurities, reagents, plasma components, degradation products, or other substances. Figure 3 shows a typical UV-Vis HPLC-derived chromatogram of 1,4-dithiothreitol (DTT), metaphosphoric acid (MPA), and a mixture of these with plasma and vitamin C standard. DTT and MPA do not interfere with the peak of vitamin C. Endogenous levels of vitamin C were detected in vitamin C-undoped plasma samples at physiological concentrations (approximately 10–20 μg/mL). The peak for vitamin C obtained in vitamin C-doped plasma samples is not well defined due to the continuous ascorbic/dehydroascorbic acid (DHA) equilibrium, but it should be quantified considering peak height.

### 2.3. Pharmacokinetic Parameters of Vitamin C After Its Oral Administration in a Reversible Model of Malabsorption

Based on our previous histological studies, we selected specific time points (1, 3, and 22 days post-administration) to monitor the intestinal damage and recovery process. Day 1 represents the initial damage phase, day 3 corresponds to the peak damage with early regeneration, and day 22 represents the near-complete recovery phase [33]. These time points provide a comprehensive profile of the intestinal injury-regeneration cycle induced by ebulin-f.

Previous histological studies show that sublethal ebulin-f doses cause maximum intestinal damage at 24 h, with full recovery by 22 days. The parenteral administration of different doses of Sambucus RILs causes a decrease in weight in the animals because of the injuries caused [33]. In the present study, the individual weights of all animals (control or treatment) were recorded. The mean weight reduction (±standard deviation) calculated with respect to time 0 at 1, 3, and 22 days after i.p. administration of 2.5 mg/kg of ebulin-f was 91.83 ± 0.35%, 84.77 ± 0.57%, 84.51 ± 0.88%, and 99.89 ± 0.18%, respectively.

A single dose of 100 mg/kg of vitamin C was administered orally to the four groups, and the corresponding plasma levels were obtained at 30, 80, 120, 180, and 1440 min after administration, according to the protocol described in the Materials and Methods section (Figure 4). These sampling time points were selected based on previously published vitamin C pharmacokinetic studies in mice [48,49] and pilot experiments that showed significant changes in vitamin C plasma concentrations at these intervals.

The pharmacokinetic analysis of vitamin C plasma concentration versus time data was performed using the SIMFIT-EXFIT package. Several models were evaluated, including one- and two-compartment models with first-order or zero-order absorption. Model selection was based on Akaike Information Criterion (AIC) values, residual analysis, and correlation of fitted curves. The two-compartment model with first-order absorption provided the best fit (lowest AIC values and most random distribution of residuals), which is consistent with the known disposition characteristics of vitamin C in mammalian systems.

The main pharmacokinetic parameters were estimated from the vitamin C plasma concentration vs. time plots previously fitted to a two-compartment model generated with the SIMFIT-EXFIT package (Table 2).

Maximum absorption time was always the same, without significant variation. Therefore, it can be deduced that there are no significant changes in the mechanism of vitamin C absorption of the maximum profiles generated in the treated animals, but there are significant changes in their kinetics.

If we look at the area under the curve (AUC 0–1440 min) of the 1-day group, it was approximately four times higher than that of the control group. On the other hand, the AUC of the 3-day group showed a significant increase compared to the 1-day group and was approximately eight times higher than that of the control group (*p* < 0.05). Finally, in the 22-day group, a significant decrease in the AUC value could be observed with respect to the 3-day group, with no statistically significant differences observed with respect to the 1-day group. It should be noted that 22 days after i.p. administration of sublethal doses of ebulin-f, the absorption values of vitamin C are still much higher than those obtained in the control group (*p* < 0.05). This may be due to the fact that the villi are already regenerated but do not yet have the structure to coat the intestine with the physiological framework as they do in an untreated individual.

It is important to note that the absorption constant had significant variations (*p* < 0.05) due to the quantitative variation of the intestinal mechanism of vitamin C absorption. In this sense, we can affirm that regeneration of the intestinal architecture promoted by ebulin-f facilitated and increased intestinal absorption.

No significant differences were observed in the elimination constants of the groups tested. This fact, together with the evaluation of the plasma concentration profiles, which tend to converge to a basal point not reached, indicates the saturation of the elimination mechanism. This also makes possible the increase in plasma concentration since the same amount of vitamin is eliminated in all the groups studied due to the saturation of the elimination system, which is unable to increase its working speed.

Comparison of the A values between the untreated group of animals and the 1-day group showed a significant increase in A values. The comparison between the 3-day group and the 1-day group also showed a significant increase in the A value. Finally, in the 22-day group, a significant decrease in the A value was observed compared to the 3-day group, although it was still higher than the value obtained by the control group.

## 3. Discussion

A comprehensive analysis of the data obtained demonstrates that the damage to the intestinal mucosa caused by the administration of sublethal doses of ebulin-f is reversible. Vitamin C absorption is increased the day after the i.p. bolus of the toxin and reaches a maximum at 3 days. Twenty-two days after administration, values are lower, tending toward those of control animals (Figure 4). According to the histological studies previously published by Jimenez et al. [35], the crypts of the small intestine would be almost recovered at that time.

The observed changes in vitamin C absorption kinetics provide valuable insights into the intestinal damage–regeneration cycle induced by ebulin-f. The significantly increased absorption observed at day 3 post-treatment is particularly noteworthy and could be explained by several possible mechanisms: (1) disruption of the intestinal barrier allowing increased paracellular transport; (2) compensatory upregulation of vitamin C transporters in remaining viable enterocytes; (3) altered redox environment in the intestinal lumen driving the conversion of ascorbic acid to dehydroascorbic acid (DHA), which utilizes different transport pathways; or (4) changes in the intestinal surface area available for absorption due to the dynamic injury–regeneration process.

The manner in which maximum absorption was observed 3 days after ebulin-f administration is likely to be due to the fact that the receptors that facilitate the diffusion of DHA are located at the level of the crypts of the small intestine. Ascorbic acid or ASC crosses the cell membrane in negligible amounts because of its anionic charge. However, in the intestine, due to the intrinsic pH being close to 5, the balance tends toward the ionized form DHA, so it may be a transport system to be taken into account, but its role is not known for certain yet [39]. Comparative studies of ascorbic acid administration and its less active isomer erythorbic acid show identical peak plasma concentrations [50,51] although it is known that erythorbic acid is not absorbed via transport facilitated by sodium-ascorbate co-transporters (SVCT1) [52]. Nonetheless, there is doubt as to whether passive transport is a mechanism of absorption of these acids.

Our findings on increased vitamin C absorption during intestinal damage are supported by recent research suggesting that GLUT transporters, particularly GLUT2 and GLUT8, may be upregulated during intestinal inflammation [46,47]. We hypothesize that the disruption of the intestinal epithelium by ebulin-f may trigger compensatory mechanisms in the remaining viable enterocytes, potentially including increased expression of both SVCT and GLUT transporters. Further studies using immunohistochemistry or PCR analysis of these transporters during the intestinal regeneration process would be valuable to confirm this hypothesis.

Sublethal i.p. doses of ebulin-f alter vitamin C absorption without affecting renal or hepatic elimination. Damage from i.p. administration of sublethal ebulin-f doses likely increases free radical concentrations in the intestinal lumen, resembling conditions in malabsorption syndromes. Thus, in animals not treated with ebulin-f, there is a balance between the oxidized and reduced forms of the vitamin, so that absorption is more prolonged in time, with active transport (saturable at high concentrations) being predominant. However, in the treated animals, the intestinal environment probably resulted in most of the vitamin being found as DHA, and thus absorption was mostly diffusional.

The enterocyte basolateral membrane has GLUT2 and GLUT8 facilitative transporters and also SVCT1 and SVCT2 active transporters [38,42] that lead to the release of ascorbic acid into the intercellular space, but its mechanism of entry into the bloodstream is still unknown. Undiscovered transporters or channels may facilitate vitamin C flux into the bloodstream. One option is the volume-sensitive anion channels in the basolateral membranes [53].

As for the excretion of vitamin C, it is a highly hydrophilic compound, and therefore, its main entry and exit system is through the kidneys. Due to its size and dissolution capacity, the ASC form filters freely by means of hydrostatic pressure through the pores of the renal corpuscle. In primary urine, due to its pH of 5, the balance of ASC tends to be 1500 times more concentrated than in the blood of the nutrient reuptake capillaries located at the proximal convoluted tubule [50]. Despite the high gradient, the absorption of ASC is negligible due to its high water solubility. In the proximal convoluted tubule, there are few hydrophilic bridges between the epithelial cells lining the tubule lumen. The proximal convoluted tubule has SLGT2 which reabsorbs about 90% of the glucose, while the other 10% is reabsorbed in the descending tubule by means of SLGT1. ASC is reported to be reabsorbed via active reuptake in the proximal renal tubules by the SVCT1 transporter [54,55]. This is a saturable active transporter, so if concentration of ASC/DHA is higher than can be reabsorbed, they will be eliminated unchanged in the urine.

The lack of significant changes in vitamin C elimination constants across all experimental groups, despite dramatic differences in absorption, suggests that renal elimination pathways were saturated at the administered dose level. This observation is consistent with previous pharmacokinetic studies of vitamin C that demonstrate saturable reabsorption mechanisms in the kidney [51,52]. The saturation of elimination pathways contributes to the sustained elevated plasma concentrations observed in our study and highlights the importance of absorption changes in determining the overall vitamin C plasma profile.

In the reversible model of malabsorption generated in mice by ebulin-f proposed by us, no changes in the level of vitamin C elimination were observed due to the collapse of the elimination systems from the beginning of the study.

Ebulin-f demonstrates higher affinity and targeting activity toward transit amplifying compartment (TAC) cells, the secondary pluripotent cells. This mechanism of action and its effect are similar to those of nigrin-b [36]. Damage to the small intestine begins on day zero after inoculation, leading to incipient destruction of the epithelial lining. At 24 h, in the small intestine, the crypts of Lieberkühn show apoptotic morphology and an atrophied profile of many of the intestinal villi. Due to the weakening of the supporting structure, villi are torn away and swept through the intestinal tract, leading to their disappearance. Likewise, the literature describes that mice, after 3 days from inoculation with nigrin-b i.p., are able to regenerate the villi of the first portion of the small intestine with high mitotic activity [37,48]. It should be noted that the model proposed concerns the pathologies most affected in malabsorption syndrome, such as Crohn’s disease and other inflammatory bowel diseases.

This reversible model of malabsorption offers several advantages for translational research on inflammatory bowel diseases. First, it provides a controlled and predictable timeline of intestinal damage and recovery, allowing precise intervention studies at specific disease stages. Second, the ability to monitor disease progression through vitamin C absorption provides a non-terminal assessment method, reducing the number of animals needed. Third, the reversible nature of the model allows longitudinal studies and reduces overall animal usage compared to terminal models. These characteristics make this model particularly valuable for evaluating therapeutic interventions for conditions like Crohn’s disease, where cycles of damage and repair are key features of the disease pathophysiology.

## 4. Conclusions

In this study, we developed and characterized a reversible model of intestinal malabsorption using sublethal doses of ebulin-f. This model demonstrates a predictable timeline of intestinal damage and regeneration that can be monitored non-invasively through plasma vitamin C levels. Key findings include the following:

(1)A validated analytical HPLC–UV/Vis method capable of accurately quantifying vitamin C in small plasma samples within a range of 6–60 µg/mL, with detection and quantification limits of 0.015 and 0.456 µg/mL, respectively.(2)Administration of a single intraperitoneal dose of 2.5 mg/kg ebulin-f induces significant but reversible intestinal damage, with peak alterations at day 3 and substantial recovery by day 22.(3)Vitamin C absorption is significantly increased during intestinal damage (up to eightfold higher AUC at day 3), without affecting elimination kinetics, suggesting that changes in absorption are directly related to intestinal epithelial damage and regeneration.

This model provides a valuable tool for investigating intestinal malabsorption syndromes and testing potential therapeutic interventions for inflammatory bowel diseases, while adhering to the 3Rs principles (replacement, reduction, refinement) in animal research.

## 5. Materials and Methods

### 5.1. Obtaining Ebulin-f from S. ebulus L. Fruits

For the isolation and characterization of ebulin-f, we used a modified version of the method previously established by our research group [30,49]. The key modifications included optimized extraction buffer composition, improved purification through a two-step chromatography process, and enhanced protein characterization methods.

Briefly, green fruit was collected in Escalona (Toledo, Spain) in June and July. Subsequently, we froze the fruit at −24 °C to maintain it as fresh until the date of ebulin-f extraction. The first step consisted of thawing and crushing the green fruit. Then, 200 g of powdered material was extracted and mixed with 800 mL of 140 mM NaCl/5 mM sodium phosphate (pH 7.5) for 12 h at 4 °C. The extract was first clarified through a cellulose lab filter, then filtered through a triple nylon mesh and centrifuged at 7500× *g* for 45 min at 4 °C. The supernatant was collected, and a second centrifugation was performed at 7500× *g* for 30 min at 4 °C.

An affinity chromatography column (AT-Sepharose 6B) was equilibrated at 2–4 °C with 280 mM NaCl/5 mM sodium phosphate buffer. Elution was performed at 4 °C with three volumes of clean-up buffer, followed by elution at 25 °C with the mobile phase doped with 120 mM lactose to block protein affinity. Elution aliquots were taken and measured using a spectrophotometer. Dialysis was performed at 4 °C in 2 L of type I EP water with continuous stirring and water changes every 12 h. The dialyzed mixture was concentrated using ultrafiltration discs, YM-10, in an Amicon^®^ stirred cell and then eluted through a Superdex 75^®^ equilibrated and eluted with 5 mM sodium phosphate buffer/0.4 M NaCl (pH 7.5). The first peak corresponds to SELfd and the second to ebulin-f. Aliquots corresponding to ebulin-f were pooled, dialyzed again with Milli-Q^®^, and finally frozen at −18 °C.

### 5.2. Ebulin-f Characterization

The purity of ebulin-f was analyzed by SDS-PAGE (dodecylsulphate-polyacrylamide gel) in the presence and absence of beta-mercaptoethanol [56].

The concentration was determined by the classical Kalb and Bernlohr spectrophotometric method [57].

A hemagglutination method was used to study protein activity. Two successive serial dilutions of 1/2 ebulin-f were performed in a microplate. Human blood type O, Rh + was used for the study and incubated at 4 °C and 24 °C for 60 min. If hemagglutination was positive, a blood scatter was generated, and if negative, a ring of blood appeared at the bottom of the wells.

### 5.3. Experimental Animal Groups for Generation of the Reversible Model with Ebulin-f

Female Swiss white CD1 mice (10 weeks of age, 30–40 g body weight) were obtained from an accredited supplier and housed under standard laboratory conditions (12-h light/dark cycle, temperature 22 ± 2 °C, humidity 55 ± 10%, food and water ad libitum). Inclusion criteria were as follows: healthy appearance, normal activity level, normal weight for age, and no previous experimental procedures. Exclusion criteria included signs of illness, abnormal behavior, weight loss >15% prior to the experiment, or pregnancy. The study was conducted in accordance with the European Communities Council guidelines (2010/63/EU) and approved by the Ethical Committee of the Animal Research and Welfare Service of the University of Valladolid (Approval Code: 607193).

Animals were randomly assigned to four groups (*n* = 6 per group): a control group receiving PBS vehicle injection, and three ebulin-f treatment groups assessed at 1 day (24 h), 3 days (72 h), and 22 days post-administration. Each treatment group received a single 100 µL i.p. injection of ebulin-f (2.5 mg/kg) dissolved in 0.1 M phosphate-buffered saline (PBS), pH 7.4. The control group received an equal volume of PBS vehicle. Body weight was monitored daily throughout the study period to assess general health status and response to treatment.

### 5.4. Vitamin C Administration

Vitamin C was administered as a single oral bolus dose of 100 mg/kg per animal in a volume of 0.1 mL. After that, blood samples were collected by the retro-orbital method by trained and certified personnel at 30, 80, 120, 180, and 1440 min to generate a complete plasma profile of the analyte. These time points were selected based on preliminary pharmacokinetic studies that identified the absorption phase (30–120 min), distribution phase (120–180 min), and elimination phase (180–1440 min) of vitamin C in mice. Plasma was obtained by centrifuging blood samples in BD Vacutainer^®^ heparin tubes for 10 min at 9000 rpm and 4 °C.

### 5.5. Vitamin C Quantification by HPLC/UV-Vis

Vitamin C analysis was performed in accordance with the methodology proposed by Esteve et al. [58] and Sánchez-Mata et al. [59]. Briefly, 20 µL of 1,4-dithiothreitol (DTT) plus 200 µL of type I EP water was added to 100 µL of plasma. DTT stabilizes vitamin C at 4 °C for 20 min. To analyze vitamin C, 200 µL of 10% metaphosphoric acid (MPA) at 4 °C was added to the plasma, vortexed for 30 s, centrifuged at 9000 rpm for 10 min, filtered through a 0.22 µm polyvinylidene difluoride (PVDF) pore membrane, and refrigerated at 4 °C without light until further analysis by HPLC. Validation was performed according to the requirements of ICH Q2A (International Conference on Harmonization). Linearity was assayed at 6, 10, 15, 30, 45, and 60 μg/mL. Repeatability was checked at low, medium, and high concentrations, i.e., 6, 30, and 60 μg/mL. All samples were prepared in triplicate from standard pharmacopeia quality (ascorbic acid USP reference standard, Merck Life Science S.L.U., Madrid, Spain).

The HPLC equipment used was a Jasco^®^ modular HPLC system (Jasco International Co. Ltd., Hachioji-shi Tokyo, Japan), with a Phenomenex^®^ Luna C18 analytical column (150.0 × 4.6 mm) and a Phenomenex^®^ C18 safeguard column (4.0 × 3.0 mm). The analysis was performed at 40 °C, using a mobile phase of H_2_SO_4_ 18 mM in type I EP water (pH 2.6), flow rate 0.9 mL/min, 15 bar pressure, UV detection at λ = 245 nm, and chromatography time of 15 min. Samples were always kept between 1 °C and 5 °C and protected from light to avoid photodegradation.

### 5.6. Statistical Analysis

The plasma concentration-time profiles were analyzed using non-compartmental and compartmental approaches. Initial non-compartmental analysis provided model-independent parameters including maximum concentration (Cmax), time to maximum concentration (Tmax), and area under the curve (AUC). For compartmental analysis, the SIMFIT-EXFIT package 6.0.24 (W.G. Bardsley, University of Manchester) was used to fit the data to different models. Model selection was based on Akaike Information Criterion (AIC) values, with the two-compartment model providing the best fit for all groups.

Normality of data was confirmed using the Shapiro–Wilk test, and homogeneity of variance was assessed using Levene’s test. Intergroup comparisons were made by one-way ANOVA followed by Tukey’s post hoc test for multiple comparisons. A *p*-value < 0.05 was considered statistically significant. All statistical analyses were performed using the same SIMFIT-EXFIT package.

## Figures and Tables

**Figure 1 toxins-17-00333-f001:**
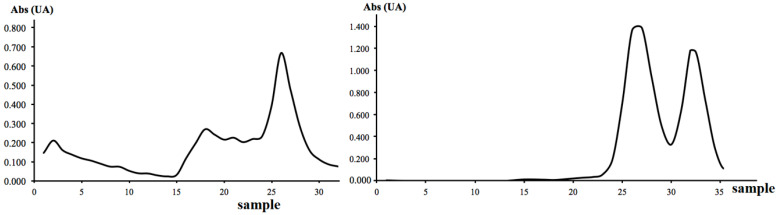
Chromatogram obtained at 280 nm by affinity chromatography with a Sepharose 6B^®^ column (**left** side). Chromatogram at 280 nm obtained by molecular size-exclusion chromatography with a Superdex 75^®^ column (**right** side).

**Figure 2 toxins-17-00333-f002:**
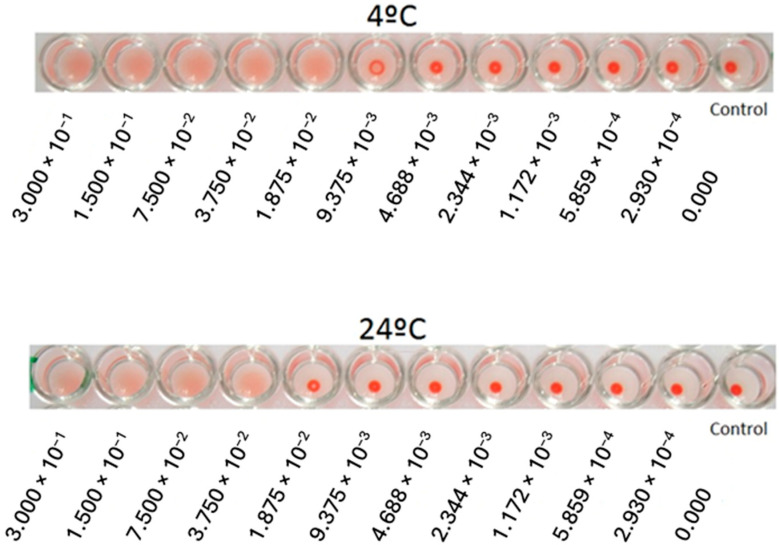
Hemagglutination activity of ebulin-f with negative controls and dilutions from 300 µg/mL to 0.2930 µg/mL.

**Figure 3 toxins-17-00333-f003:**
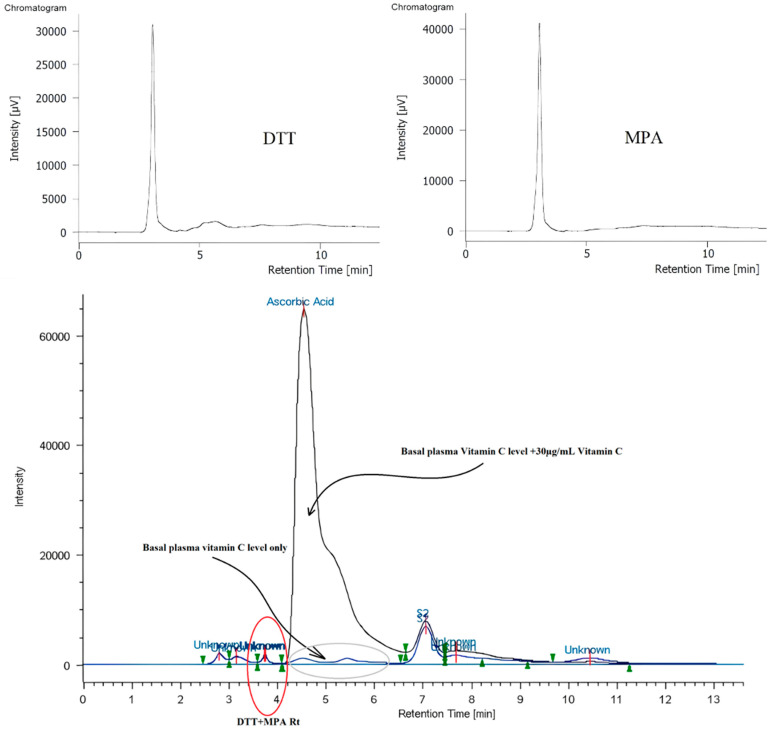
Vitamin C-DTT chromatogram (**top**-**left**), vitamin C-MPA chromatogram (**top**-**right**), and vitamin C-undoped plasma chromatogram overlaid with a vitamin C-doped plasma chromatogram (**bottom**).

**Figure 4 toxins-17-00333-f004:**
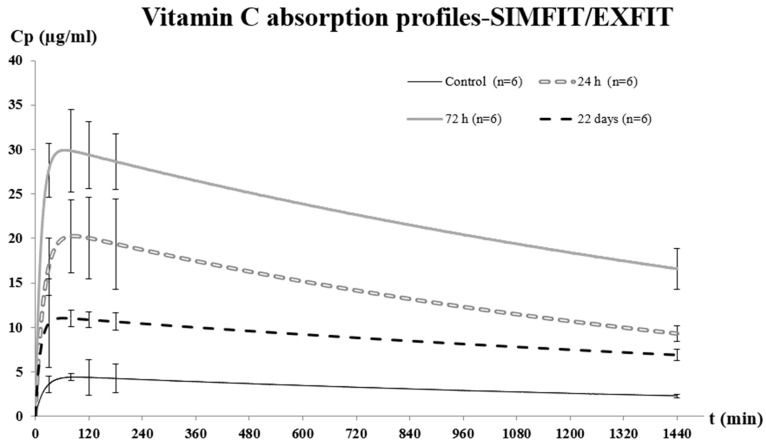
Plasma concentrations of vitamin C after oral administration of a single dose of 100 mg/kg in mice treated at different times with a sublethal dose of ebulin-f. The control group has not been treated with the RIP.

**Table 1 toxins-17-00333-t001:** Results of the main validation parameters of the method for quantification of vitamin C in plasma according to the ICH Q2A validation of analytical methods recommendations.

Method Validation. Vitamin C in Water		
Linearity test	Concentration range (µg/mL)	6, 10, 15, 130, 45, and 60
	Number of samples	3
	Calibration equation	y = 85,373.8922x + 10,905.9591
	r	0.9998
	r^2^	0.9995
Other parameters	Concentration range (µg/mL)	2, 30 and 60
	Number of samples	3
	Detection limits (µg/mL)	0.015
	Quantitation limits (µg/mL)	0.456
	Cochram test accuracy	
	Recovery % mean	data
	Student’s *t*-test accuracy	
	Repeatability (CV) (<5% ICH Q2A)	6 µg/mL, CV = 3.81%; 30 µg/mL, CV = 1.85%; 60 µg/mL, CV = 1.94%;
	Precision (<3.3% ICH Q2A)	0.1048%
	Robustness	CV < 3% with variations in temperature (±2 °C), flow rate (±0.1 mL/min), and mobile phase pH (±0.2)

**Table 2 toxins-17-00333-t002:** Vitamin C absorption constants, elimination constants, intercepts, and areas under the curve (AUC) obtained at 0, 1, 3, and 22 days after i.p. administration of 2.5 mg/kg of ebulin-f. Different symbols (*, +, $) indicate significant differences between groups (*p* < 0.05) using one-way ANOVA followed by Tukey’s post hoc test (n = 6 per group).

	Control	1 Day	3 Days	22 Days
Absorption constant (A) Confidence interval (±)	4.6 *	21.5 ^+^	31.0 ^$^	11.3 ^+,$^
	0.5	1.5	19.2	0.8
Elimination constant Confidence interval (±)	4.90 × 10^−4^	5.83 × 10^−4^	4.34 × 10^−4^	3.44 × 10^−4^
	2.16 × 10^−4^	1.23 × 10^−4^	1.31 × 10^−4^	1.07 × 10^−4^
Interception constant Confidence interval (±)	5.23 × 10^−2^	5.34 × 10^−2^	7.91 × 10^−2^	9.13 × 10^−2^
	2.56 × 10^−2^	1.84 × 10^−2^	3.57 × 10^−2^	6.55 × 10^−2^
AUC Confidence interval (±)	0.94 × 10^4^ *	3.65 × 10^4 +^	7.10 × 10^4 +,$^	3.27 × 10^4 $^
	0.37 × 10^4^	0.67 × 10^4^	1.96 × 10^4^	0.91 × 10^4^

## Data Availability

The original contributions presented in this study are included in the article. Further inquiries can be directed to the corresponding author(s).

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
