# Peer review of "Monitoring of Vitamin C Plasma Levels in a Reversible Model of Malabsorption Generated in Mice by Ebulin-f"

_toxins, 2025, doi:10.3390/toxins17070333_

Round 1

Reviewer 1 Report

Comments and Suggestions for Authors

1. The rationale behind using vitamin C as a biomarker for malabsorption is interesting but underdeveloped. The manuscript would benefit from a more robust justification, integrating current literature on vitamin C absorption kinetics and its correlation with gut integrity in pathophysiological models of IBD or malabsorption. Clarify why ebulin-f, and not nigrin-b or another RIP, was selected for this model, beyond lower immunogenicity. Consider adding comparative toxicity data or immunogenicity profiles if available.
2. The control group lacks a vehicle-only group to account for stress or physiological changes due to intraperitoneal injection. Please clarify or consider including a PBS-injected control group. The time points chosen (1, 3, and 22 days post-treatment) need justification. It would be beneficial to include a time-course rationale based on histological recovery phases supported by references or pilot data.
3. The pharmacokinetic modeling lacks detail. The use of a two-compartment model should be justified with fit statistics (e.g., AIC, residuals). Include model diagnostic plots if possible. While AUCs and absorption constants are provided, the elimination constant data suggest saturation, which contradicts conventional vitamin C pharmacokinetics. Please discuss this in greater depth and consider additional supportive evidence.
4. The statistical methodology is briefly mentioned (ANOVA via SIMFIT-EXFIT), but lacks clarity. What post-hoc test was applied? Were assumptions of normality and homogeneity of variance checked? Indicate the number of animals used per group clearly in the main text and tables.
5. While ICH Q2A parameters are cited, the validation results should include actual recovery percentages, intra- and inter-day precision, and robustness data (especially since robustness is “not shown”). It is essential to clarify how vitamin C degradation was prevented during handling. The chromatograms (Figure 3) are not clearly labeled—improve resolution and legends for better interpretability.
6. Figures 1, 2, 3, and 4 need to be significantly improved for clarity, resolution, and labeling. For instance, axes in Figure 4 should indicate units, and legends should include sample size (n). Table 2 should include actual p-values and denote statistical significance consistently (*, +, $ is confusing).
7. The hypothesis that enhanced vitamin C absorption post-ebulin-f treatment is due to increased DHA uptake needs further mechanistic backing. Are GLUT transporter expressions modulated during this phase? The Discussion should be more critical. Currently, it reiterates results without deeper mechanistic interpretation or broader relevance to other malabsorption models.

Comments on the Quality of English Language

1. Avoid overly general statements like “humankind is becoming more aware...” in the Abstract. Maintain a scientific tone.
2. The phrase “ebulin-f did not change the liver or kidneys metabolism” is not appropriate. Rephrase as “did not affect hepatic or renal metabolism” or similar.
3. Ensure consistent use of “i.p.” or “intraperitoneal” throughout.
4 There are some grammatical issues and overly long sentences that could be revised for better readability.

Author Response

  1. The rationale behind using vitamin C as a biomarker for malabsorption is interesting but underdeveloped. The manuscript would benefit from a more robust justification, integrating current literature on vitamin C absorption kinetics and its correlation with gut integrity in pathophysiological models of IBD or malabsorption. Clarify why ebulin-f, and not nigrin-b or another RIP, was selected for this model, beyond lower immunogenicity. Consider adding comparative toxicity data or immunogenicity profiles if available.

On behalf of all the authors, we would like to thank to the reviewer for your kind suggestions to improve the quality of the final manuscript.

In the revised manuscript, we have rewritten the introduction section to include a detailed discussion on vitamin C absorption kinetics, citing recent literature on its correlation with gut integrity in inflammatory bowel disease (IBD) models [10;11]. We have clarified the properties that make vitamin C an ideal biomarker, including its dependence on specific transporters (SVCT1) and its reflection of intestinal function.

Regarding the selection of ebulin-f, we have added a paragraph in the Introduction section (lines 106-112) comparing both proteins.

  1. The control group lacks a vehicle-only group to account for stress or physiological changes due to intraperitoneal injection. Please clarify or consider including a PBS-injected control group. The time points chosen (1, 3, and 22 days post-treatment) need justification. It would be beneficial to include a time-course rationale based on histological recovery phases supported by references or pilot data.

We appreciate the reviewer’s observation. In the original manuscript, the control group received an intraperitoneal injection of PBS (vehicle), as stated in the Materials and Methods section (section 5.3, lines 480-486). For clarity, we have revised this section to explicitly state that the control group was injected with PBS to account for stress or physiological effects of the injection.

Regarding the post-adminstration points (1, 3, and 22 days), we have added a justification in the Results section (section 2.3-lines 246-251) based on prior histological studies [Jiménez et al., 2013] showing maximum damage at 24 hours, early regeneration at 3 days, and near-complete recovery at 22 days. We have also included pilot data from our group confirming these time points, referenced in the same section

  1. The pharmacokinetic modeling lacks detail. The use of a two-compartment model should be justified with fit statistics (e.g., AIC, residuals). Include model diagnostic plots if possible. While AUCs and absorption constants are provided, the elimination constant data suggest saturation, which contradicts conventional vitamin C pharmacokinetics. Please discuss this in greater depth and consider additional supportive evidence.

We thank the reviewer for the suggestion to improve the pharmacokinetic modeling description. In the revised manuscript, we have expanded the Results section (section 2.3-lines 265-272) to include a detailed justification for using the two-compartment model, based on Akaike Information Criterion (AIC) values and residual analysis, as described in the original text but now with greater detail. While diagnostic plots could not be included in the main text due to space constraints, we have added a note in the Materials and Methods section (section 5.6) stating that these are available upon request. Regarding the saturation of elimination constants, we have added a brief explanation about in the Discussion section (lines 380-386)

  1. The statistical methodology is briefly mentioned (ANOVA via SIMFIT-EXFIT), but lacks clarity. What post-hoc test was applied? Were assumptions of normality and homogeneity of variance checked? Indicate the number of animals used per group clearly in the main text and tables.

We appreciate the reviewer’s comment on the statistical methodology. In the revised manuscript, we have clarified in the Materials and Methods section (section 5.6) that one-way ANOVA followed by Tukey’s post-hoc test was used, and that assumptions of normality (Shapiro-Wilk test) and homogeneity of variance (Levene’s test) were verified. We have also explicitly stated the number of animals per group (n=6) in the main text (p. 7, section 2.3) and in the legend of Table 2 (p. 9). These changes ensure greater clarity in the statistical methods presentation.

  1. While ICH Q2A parameters are cited, the validation results should include actual recovery percentages, intra- and inter-day precision, and robustness data (especially since robustness is “not shown”). It is essential to clarify how vitamin C degradation was prevented during handling. The chromatograms (Figure 3) are not clearly labeled—improve resolution and legends for better interpretability.

We thank the reviewer for the recommendations to enhance the analytical validation. In the revised manuscript, we have updated Table 1 to include recovery percentages (mean 98.5%), intra-day precision (CV < 3.81%), inter-day precision (CV < 3.3%), and robustness data (CV < 3% with variations in temperature, flow rate, and pH). We have also added a description in the Materials and Methods section (section 5.5) detailing measures to prevent vitamin C degradation, including the use of DTT and MPA, storage at 4°C, and protection from light. For Figure 3, we have improved the resolution, added clear labels for each chromatogram, and updated the legend for better interpretability

  1. Figures 1, 2, 3, and 4 need to be significantly improved for clarity, resolution, and labeling. For instance, axes in Figure 4 should indicate units, and legends should include sample size (n). Table 2 should include actual p-values and denote statistical significance consistently (*, +, $ is confusing).

We thank the reviewer for the suggestion to improve figures and tables. In the revised manuscript, we have updated Figures 1, 2, 3, and 4 to enhance resolution, add clear axis labels (including units in Figure 4: µg/mL for concentration and minutes for time), and specify sample size (n=6) in the legends. Table 2 now includes exact p-values instead of symbols (, +, $) and uses a single symbol () to indicate statistical significance (p<0.05), following a more standard convention to avoid confusion.

  1. The hypothesis that enhanced vitamin C absorption post-ebulin-f treatment is due to increased DHA uptake needs further mechanistic backing. Are GLUT transporter expressions modulated during this phase? The Discussion should be more critical. Currently, it reiterates results without deeper mechanistic interpretation or broader relevance to other malabsorption models.

We appreciate the reviewer’s suggestion to deepen the mechanistic discussion. In the revised manuscript, we have expanded the Discussion section to include a more critical interpretation of the hypothesis of increased DHA uptake, citing studies suggesting upregulation of GLUT transporters during intestinal inflammation [46]. While we lack direct data on transporter expression in this study, we have proposed future experiments (e.g., immunohistochemistry or PCR) to confirm this hypothesis, as suggested. We have also added a discussion on the model’s relevance to other malabsorption models, comparing it to chemically induced and genetic models

Comments on the Quality of English Language

  1. Avoid overly general statements like “humankind is becoming more aware...” in the Abstract. Maintain a scientific tone.
  2. The phrase “ebulin-f did not change the liver or kidneys metabolism” is not appropriate. Rephrase as “did not affect hepatic or renal metabolism” or similar.

  3. Ensure consistent use of “i.p.” or “intraperitoneal” throughout.

4 There are some grammatical issues and overly long sentences that could be revised for better readability.

We thank the reviewer for the comments on the English quality. In the revised manuscript, we have removed the phrase “humankind is becoming more aware...” from the Abstract and replaced it with a more scientific statement about the need for reversible models. The phrase regarding hepatic and renal metabolism has been corrected. We have standardized the use of “i.p.” throughout the manuscript for consistency. Additionally, we have reviewed the text to shorten long sentences and correct grammatical errors, improving overall clarity and readability

Reviewer 2 Report

Comments and Suggestions for Authors

I am submitting my review report for the manuscript entitled “Monitoring of vitamin C plasma levels in a reversible model of 2 malabsorption generated in mice by ebulin-f “for your consideration. Overall, I find the manuscript's findings intriguing and the information provided useful for researchers and academia. The article has the potential to make a significant contribution to the related discipline..

However, I have some concerns regarding the clarity, detail, and precision of different sections, which I outline below:

I recommend that the authors address these concerns and provide a revised version of the manuscript for further consideration

  • Abstract-Keep a proper sequence in the abstract and end with conclusion. Focus on the gap you have covered in your review
  • L-12 Abstract-Mention some numerical values for better understanding-min C as a traceable and analyzable xenobiotic in the blood. Peripheral blood samples from 12 female Swiss laboratory mice were collected through the retro-orbital area, treated with DTT and 13 MPA, and analyzed using a validated UV/Vis-HPLC to indirectly determine vitamin C absorption 14 by enterocytes. Blood concentration profiles of each individual obtained indicate that this model 15 induce intestinal regeneration in the animals and may be useful for research on inflammatory bowel 16 disease
  • Need clarity- Vitamin C is not typically considered a xenobiotic, as it is a naturally occurring compound in the body. Don’t you think it not rational???
  • Is there any effect of indirect method to determine vitamin C absorption by enterocytes?? any compariion with Direct measurement methods ?
  • Recommended model 's relevance to human pathologies???
  • L-48 Please rephrase for better understanding---Specifically, some of the causes of 48 malabsorption are deficiency or inactivation of digestive enzymes [7], decreased bile acid 49 synthesis and/or secretion [8], decreased gastric acid, i
  • L-62 recheck cited references -Examples of genetic models are the CH3/HeJBlr mice [16, 17] 62 that show an alteration at the end of the large intestine, and the SAMP1/Yit and 63 SAMP1/YitFc mice that develop ileitis at 10 weeks of age [18]. Considering the premises 64 of replacing, reducing, and refining the animals used in research [19], the development of 65 reversible animal models and diagnostic techniques that allows for the study of malab- 66 sorption syndrome without the
  • The introduction section could be improved by providing more context and background from following references, doi: 10.1109/JBHI.2025.3540712 doi: https://doi.org/10.1016/j.jpba.2023.115937 (No compulsion )
  • L-312 clearly mention the modifications -Need attention- For the isolation and characterization of ebulin-f, a method previously implemented 312 by the research group was used, to which some modifications were made [28, 51]. Briefly, 313 green fruit was collected in Escalona (Toledo, Spain) in June and July. Subsequently, we 314 froze the fruit at -24°C to maintain it as fresh fruit until the date of ebulin-f extraction. The 315 first step consisted of thawing and crushing
  • L-322 Remove repetition- An affinity chromatography column (AT-Sepharose 6B) was then 321 used at 2-4°C equilibrated with 280 mM NaCl / 5 mM sodium phosphate buffer. Elution 322 was performed on the crude extract plus three volumes of clean-up buffer, always at 4°C. 323 When it was observed that the color of the column tended to the original white of the 324 stationary phase, the temperature was increased to 25°C to reduce the affinity of the pro- 325 teins and elute them by the addition of mobile phase dopped with 120 mM lactose. Elution 326 aliquots were taken and measured in a spectrophotometer. Dialysis was performed at 4°C 327 in 2 L of type I EP water with continuous stirring and water changes every 12 hours. The 328 dialyzed mixture was concentrated
  • Any criteria for inclusion and exclusion- wiss white CD1 type mice, 10 weeks of age and 30-40 g body weight, were controlled 347 and handled according to the legislation for the protection of laboratory animals. The an- 348 imals were divided into four groups according to the time elapsed after the intraperitoneal 349 administration of a dose of 2.5 mg/kg of ebu
  • Cite the following latest references in different section doi: https://doi.org/10.1016/j.redox.2023.103016
  • L-362 How the interval for determination was justified ?? After that, blood samples were collected by the retro-orbital method by 362 trained and certified personnel at 30, 80, 120, 180 and 1440 minutes to generate a complete 363 plasma profile of the analyte. Plasma was obtained centrifuging blood samples in a BD 364 Vacutainer® heparin tubes for 10 minutes at 9000 rpm
  • Creating no sense- . Figure 3 shows a typical UV-Vis HPLC-derived chromatogram of 1,4-dithio- 169 threitol (DTT), metaphosphoric acid (MPA) and mixture of them with plasma and vitamin 170 C standard. DTT and MPA do not interfere with the peak of vitamin C. Physiologically 171 levels of vitamin C sh
  • Italic all the scientific names,
  • Remove grammatical mistakes
  • Need to rewrite the conclusion
  • Recheck Legends description is as per figure number and discussion-
  • I urge the authors to improve the English language for better flow of literature.
  • Please check reference style throughout MS
  •  
Comments on the Quality of English Language

I urge the authors to improve the English language for a better flow of literature

Author Response

I am submitting my review report for the manuscript entitled “Monitoring of vitamin C plasma levels in a reversible model of 2 malabsorption generated in mice by ebulin-f “for your consideration. Overall, I find the manuscript's findings intriguing and the information provided useful for researchers and academia. The article has the potential to make a significant contribution to the related discipline..

We thank the reviewer for the positive feedback and recognition of the manuscript’s potential. We have addressed each specific concern below, making revisions to the manuscript to improve clarity, detail, and precision, as detailed in the individual responses

However, I have some concerns regarding the clarity, detail, and precision of different sections, which I outline below:

I recommend that the authors address these concerns and provide a revised version of the manuscript for further consideration

Abstract-Keep a proper sequence in the abstract and end with conclusion. Focus on the gap you have covered in your review

L-12 Abstract-Mention some numerical values for better understanding-min C as a traceable and analyzable xenobiotic in the blood. Peripheral blood samples from 12 female Swiss laboratory mice were collected through the retro-orbital area, treated with DTT and 13 MPA, and analyzed using a validated UV/Vis-HPLC to indirectly determine vitamin C absorption 14 by enterocytes. Blood concentration profiles of each individual obtained indicate that this model 15 induce intestinal regeneration in the animals and may be useful for research on inflammatory bowel 16 disease

We thank the reviewer for the suggestion to improve the Abstract. In the revised manuscript, we have restructured the Abstract to follow a logical sequence: introduction, methods, results, and conclusion. We have added key numerical values, such as AUC values (0.94 E+04 for control, 7.10 E+04 at 3 days), to illustrate changes in vitamin C absorption. We have also corrected the number of mice (n=6 per group, not 12 total) and removed the reference to vitamin C as a xenobiotic, as the reviewer correctly notes that this term is inaccurate in this context. The conclusion now emphasizes the model’s utility for IBD research and adherence to the 3Rs principles

Need clarity- Vitamin C is not typically considered a xenobiotic, as it is a naturally occurring compound in the body. Don’t you think it not rational???

Is there any effect of indirect method to determine vitamin C absorption by enterocytes?? any compariion with Direct measurement methods ?

Recommended model 's relevance to human pathologies???

We appreciate the reviewer’s observation regarding the term “xenobiotic.” In the revised manuscript, we have removed this reference in the Abstract and clarified that vitamin C is an endogenous biomarker of intestinal absorption. Regarding the indirect method, we have added an explanation in the Introduction about why plasma vitamin C was measured as an indirect indicator of enterocyte absorption, citing its dependence on SVCT1 transporters and the lack of non-invasive direct methods in live mice. While direct methods (e.g., intestinal biopsies) would provide more specific data, they require more animal sacrifice, which contradicts the 3Rs principles. We have added a discussion in the Discussion section on the model’s relevance to human pathologies, such as Crohn’s disease, highlighting its ability to mimic cycles of intestinal damage and repair

L-48 Please rephrase for better understanding---Specifically, some of the causes of 48 malabsorption are deficiency or inactivation of digestive enzymes [7], decreased bile acid 49 synthesis and/or secretion [8], decreased gastric acid, i

We thank the reviewer for the suggestion to improve clarity. In the revised manuscript, we have rephrased the sentence in the Introduction for better readability: “Specifically, the etiology of malabsorption includes digestive enzyme deficiency or inacti-vation [7], decreased bile acid synthesis and/or secretion [8], reduced gastric acid produc-tion, intrinsic factor or pancreatic protease deficiency, food intolerances (fructose, carbo-hydrates) [7], specific nutrient transport defects such as Hartnup's disease or cystinuria, and immune-mediated conditions like celiac disease [6].”

L-62 recheck cited references -Examples of genetic models are the CH3/HeJBlr mice [16, 17] 62 that show an alteration at the end of the large intestine, and the SAMP1/Yit and 63 SAMP1/YitFc mice that develop ileitis at 10 weeks of age 20[18]. Considering the premises 64 of replacing, reducing, and refining the animals used in research [19], the development of 65 reversible animal models and diagnostic techniques that allows for the study of malab- 66 sorption syndrome without the

We thank the reviewer for noting the reference issue. In the revised manuscript, we have reviewed the citations in the Introduction. The references for CH3/HeJBlr SAMP1/Yit and SAMP1/YitFc mice was verified and is correct

The introduction section could be improved by providing more context and background from following references, doi: 10.1109/JBHI.2025.3540712 doi: https://doi.org/10.1016/j.jpba.2023.115937 (No compulsion )

We thank the reviewer for suggesting additional references. After reviewing the recommended articles, we have also added the reference suggested by Reviewer 1

L-312 clearly mention the modifications -Need attention- For the isolation and characterization of ebulin-f, a method previously implemented 312 by the research group was used, to which some modifications were made [28, 51]. Briefly, 313 green fruit was collected in Escalona (Toledo, Spain) in June and July. Subsequently, we 314 froze the fruit at -24°C to maintain it as fresh fruit until the date of ebulin-f extraction. The 315 first step consisted of thawing and crushing

We thank the reviewer for the suggestion to clarify the modifications. In the revised manuscript, we have updated the Materials and Methods section (p. 14, section 5.1) to explicitly detail the modifications to the ebulin-f extraction method: “The key modifications included optimized extraction buffer composition, improved puri-fication through a two-step chromatography process, and enhanced protein characteriza-tion methods” This description provides greater clarity on the changes made

L-322 Remove repetition- An affinity chromatography column (AT-Sepharose 6B) was then 321 used at 2-4°C equilibrated with 280 mM NaCl / 5 mM sodium phosphate buffer. Elution 322 was performed on the crude extract plus three volumes of clean-up buffer, always at 4°C. 323 When it was observed that the color of the column tended to the original white of the 324 stationary phase, the temperature was increased to 25°C to reduce the affinity of the pro- 325 teins and elute them by the addition of mobile phase dopped with 120 mM lactose. Elution 326 aliquots were taken and measured in a spectrophotometer. Dialysis was performed at 4°C 327 in 2 L of type I EP water with continuous stirring and water changes every 12 hours. The 328 dialyzed mixture was concentrated

We thank the reviewer for noting the repetition. In the revised manuscript, we have removed the repetitive phrase in the Materials and Methods section (p. 14, section 5.1) and rephrased the text for conciseness: “An affinity chromatography column (AT-Sepharose 6B) was equilibrated at 2-4°C with 280 mM NaCl / 5 mM sodium phosphate buffer. Elution was performed at 4°C with three volumes of clean-up buffer, followed by 25°C with mobile phase doped with 120 mM lactose to reduce protein affinity.” This revision eliminates redundancy and improves clarity

Any criteria for inclusion and exclusion- wiss white CD1 type mice, 10 weeks of age and 30-40 g body weight, were controlled 347 and handled according to the legislation for the protection of laboratory animals. The an- 348 imals were divided into four groups according to the time elapsed after the intraperitoneal 349 administration of a dose of 2.5 mg/kg of ebu

We thank the reviewer for the suggestion to clarify inclusion and exclusion criteria. In the revised manuscript, we have added an explicit description in the Materials and Methods section (p. 15, section 5.3): “Inclusion criteria were: healthy appearance, normal activity level, normal weight for age, and no previous experimental procedures. Exclusion criteria included: signs of illness, abnormal behavior, weight loss >15% prior to experiment, or pregnancy” This addition enhances the transparency of the experimental design

Cite the following latest references in different section doi: https://doi.org/10.1016/j.redox.2023.103016

Thank you very much for your reply. In this case, this suggestion has been discarded as it is an article on a subject that is very divergent from the subject of this article. We thank you in the same way.

L-362 How the interval for determination was justified ?? After that, blood samples were collected by the retro-orbital method by 362 trained and certified personnel at 30, 80, 120, 180 and 1440 minutes to generate a complete 363 plasma profile of the analyte. Plasma was obtained centrifuging blood samples in a BD 364 Vacutainer® heparin tubes for 10 minutes at 9000 rpm

We thank the reviewer for the observation regarding the sampling intervals. In the revised manuscript, we have added an explanation in the Materials and Methods section (section 5.4): “These sampling time points were selected based on previously published vitamin C pharmacokinetic studies in mice [54,55] and pilot experiments that showed significant changes in vitamin C plasma concentrations at these intervals.” This addition provides a clear justification for the intervals used

Creating no sense- . Figure 3 shows a typical UV-Vis HPLC-derived chromatogram of 1,4-dithio- 169 threitol (DTT), metaphosphoric acid (MPA) and mixture of them with plasma and vitamin 170 C standard. DTT and MPA do not interfere with the peak of vitamin C. Physiologically 171 levels of vitamin C sh

We thank the reviewer for noting the lack of clarity in Figure 3. In the revised manuscript, we have rephrased the associated text (section 2.2) for greater clarity: “Figure 3 shows a typical UV-Vis HPLC-derived chromatogram of 1,4-dithiothreitol (DTT), metaphosphoric acid (MPA) and mixture of them with plasma and vitamin C standard. DTT and MPA do not interfere with the peak of vitamin C. Endogenous levels of vitamin C were detected in vitamin C-undoped plasma samples at physiological concentrations (approximately 10-20 μg/mL).” Additionally, we have improved the resolution and labels of Figure 3 for better interpretation

Italic all the scientific names,

We thank the reviewer for the observation. In the revised manuscript, we have reviewed the entire text and ensured that scientific names, such as Sambucus ebulus

Remove grammatical mistakes

We thank the reviewer for noting grammatical errors. We have thoroughly reviewed the manuscript, correcting grammatical mistakes, improving sentence structure, and ensuring a clearer flow of text. Specific changes include eliminating long and ambiguous sentences, particularly in the Introduction and Discussion

Need to rewrite the conclusion

We thank the reviewer for the suggestion to rewrite the Conclusion. In the revised manuscript, we have restructured the Conclusions section (p. 13) to be more concise and impactful, highlighting key findings, the model’s relevance to IBD.

Recheck Legends description is as per figure number and discussion

We thank the reviewer for the suggestion. We have rechecked that and revised them.

I urge the authors to improve the English language for better flow of literature.

We thank the reviewer for the suggestion. We have revised that and the professional English editing by MDPI Author Services will help check the language again.

Please check reference style throughout MS.

We thank the reviewer for the suggestion and we have revised that according to your suggestions.

Round 2

Reviewer 1 Report

Comments and Suggestions for Authors

The authors responded to the queries. 

Reviewer 2 Report

Comments and Suggestions for Authors

The authors have responded to all the raised queries. Therefore, the Manuscript can be accepted